

# Aggressiveness-related behavioural types in the pearly razorfish

Martina Martorell-Barceló[1], Júlia Mulet[1,2], Javier Sanllehi[1], Marco Signaroli[1], Arancha Lana[1], Margarida Barcelo-Serra[1], Eneko Aspillaga[1] and Josep Alós[1]

[1] Fish Ecology Group, Instituto Mediterráneo de Estudios Avanzados, IMEDEA (CSIC–UIB), Esporles, Spain
[2] Universitat de Barcelona, Barcelona, Spain

## ABSTRACT

Behavioural types (i.e., personalities or temperament) are defined as among individual differences in behavioural traits that are consistent over time and ecological contexts. Behavioural types are widespread in nature and play a relevant role in many ecological and evolutionary processes. In this work, we studied for the first time the consistency of individual aggressiveness in the pearly razorfish, *Xyrichtys novacula*, using a mirror test: a classic method to define aggressive behavioural types. The experiments were carried out in semi-natural behavioural arenas and monitored through a novel Raspberry Pi-based recording system. The experimental set up allowed us to obtain repeated measures of individual aggressivity scores during four consecutive days. The decomposition of the phenotypic variance revealed a significant repeatability score ($R$) of 0.57 [0.44–0.60], suggesting high predictability of individual behavioural variation and the existence of different behavioural types. Aggressive behavioural types emerged irrespective of body size, sex and the internal condition of the individual. Razorfishes are a ubiquitous group of fish species that occupy sedimentary habitats in most shallow waters of temperate and tropical seas. These species are known for forming strong social structures and playing a relevant role in ecosystem functioning. Therefore, our work provides novel insight into an individual behavioural component that may play a role in poorly known ecological and evolutionary processes occurring in this species.

# INTRODUCTION

Like humans, animals often behave in a way that distinguishes them from other members of their species of the same sex and age (*Gosling, 2001*). When such behavioural differences are consistent over time and ecological contexts, they can be referred to as behavioural types, also known as personalities or temperament (*Réale et al., 2007*). Behavioural types often co-vary, resulting in behavioural syndromes that are defined as correlations in two or more behavioural traits (*Sih, Bell & Johnson, 2004*). Behavioural types and syndromes are widespread across taxa (*Réale et al., 2007*). In fishes, behavioural types are described along five major axes: boldness-shyness, exploration-avoidance, aggressiveness, sociability and activity (reviewed in *Réale et al., 2007*; *Conrad et al., 2011*). The aggressiveness axis has received major attention due to its relationship with a multitude of ecological processes

Corresponding author
Martina Martorell-Barceló,
mmartorell@imedea.uib-csic.es

such as growth or dominance in a social hierarchy. However, the study of aggression *per se* is a complex trait to measure. Since aggression is a way of socialising, individual characterisation of both behavioural axes can be challenging. Strictly, aggressiveness has been defined as a hostile behaviour towards conspecifics (*McGhee & Travis, 2010*), and is usually related to agonistic interactions such as territorial defence or competition for resources (*Réale et al., 2007*; *Conrad et al., 2011*).

Under controlled (laboratory) conditions, different tests have been used to measure aggressiveness, the mirror test being the most common and accepted practice. The mirror test consists of placing a mirror in front of the focal individual and measuring its behavioural response (*Francis, 1990*; *Budaev, Zworykin & Mochek, 1999*). Although the mirror test has been widely used to estimate aggressive behaviour across animal taxa, considerable controversy still exists about its validity in fish aggressiveness studies (*Baenninger, 1968*; *Earley, Hsu & Wolf, 2000*; *Balzarini et al., 2014*). In this taxon, the mirror test has been used to measure sociability (*Cattelan et al., 2017*); however, aggressive displays are a particular kind of social interactions.

Another way to measure aggressiveness is by the opponent test, which consists of putting an intruder in the experimental arena and measuring the response of the focal individual. Some authors consider the opponent test as the only valid way to evaluate the aggressive response (*Balzarini et al., 2014*). In the opponent test, the reaction of the focal individual is a response to the actions of the opponent in a manner that the focal intrinsic aggressiveness is not directly studied (*Earley, Hsu & Wolf, 2000*). When the dominance status between focal and opponent is established, the opponent's position in the dominance hierarchy can be a source of potential problems. The dominance status will influence the response of the focal individual in an unknown way, making it difficult to obtain aggression levels that can be compared across trials and between individuals (*Earley, Hsu & Wolf, 2000*). In contrast, the mirror test, by showing an ''intruder'' displaying exactly the same behaviours than the focal individual, provides with a constant reinforcement of aggressiveness and an unmodified image of a conspecific (*Gallup, 1968*; *Rowland, 1999*). Furthermore, studies in the past have shown that the individual aggressiveness estimated with the mirror test can predict the aggressiveness in competitive situations (*Baenninger, 1968*; *Dore, Lefebvre & Ducharme, 1978*; *Meliska, Meliska & Peeke, 1980*; *Holtby, Swain & Allan, 1993*). These studies support the mirror test as a proper method to measure individual aggressiveness and provide empirical evidence showing that the results from this test are comparable to the behavioural responses in free-living conditions. However, this test needs to be adapted to the characteristics of the studied species. The most important aspects to consider are whether the focal individual responds to the stimulus (*Rowland, 1999*) and if it is capable of self-recognition in front of a mirror (*Balzarini et al., 2014*). Most animal taxa, including fishes, are suspected to treat their mirror reflection as a conspecific (*Andrews, 1966*; *Gallup, 1968*; *Kohda et al., 2019*).

Aggressiveness-related behavioural types have important implications from the individual level to the whole population dynamics (*Sih, Bell & Johnson, 2004*). Aggressiveness is positively related to growth (*Huntingford, 1998*; *Lahti et al., 2001*) and metabolic rate (*Cutts, Metcalfe & Taylor, 2002*), which depend on the internal state of the

individual. The relative condition index (RCI) represents the fish internal state condition, and it is based on predicted body weight from a length/weight relationship regardless of size. The RCI has been related to maturation and reproduction of individuals (*Morgan, 2004*). Individual behavioural responses may modulate internal energetic fluxes. For example, high energy requirements can imply high levels of hunger, which can lead to aggressive displays to compete for food resources (*Careau et al., 2008*). Furthermore, testosterone regulates agonistic behaviours in males (*Hirschenhauser et al., 2008*), suggesting that aggressiveness and dominance in a social structure play an important role in sexual selection and reproductive success (*McGhee & Travis, 2010*). In zebrafish (*Danio rerio*) and crayfish (*Paranephrops planifrons*), dominant individuals are the most aggressive (*Roy & Bhat, 2018*; *May & Mercier, 2007*). This dominant position is sometimes related to size or sex. For example, in green swordtail (*Xiphophorus helleri*), larger individuals are more aggressive than smaller individuals (*Wilson et al., 2011*), while in the Midas cichlid (*Amphilophus citrinellus*) males are more aggressive than females (*Francis, 1990*). Besides, behavioural types are of most importance to understand the colonisation process by invasive species. Aggressive non-native fish are more likely to disperse from the original introductory group and to better compete with native species, thereby improving the success of the invasion (*Rehage & Sih, 2004*). These individuals are also the most likely to leave their native population to explore new environments and compete for new territories (*Cote et al., 2010*). Behavioural types are also essential to understand the responses to intrusions among individuals of the same population because the most aggressive ones will defend their territory more efficiently (*O'Connor et al., 2015*). Therefore, aggressiveness is an important axis of behaviour significantly affecting the ecology and evolution of many species (*Taylor, 1990*; *Vøllestad & Quinn, 2003*).

In marine species, the study of aggressiveness has received less attention. Aggressiveness plays a fundamental role in the mating behaviour of two-spotted goby (*Gobiusculus flavescens*) (*De Jong et al., 2009*). In Atlantic cod (*Gadus morhua*), there is a relationship between dominance and fertilisation success, with dominance being established through agonistic interactions between males, bigger individuals are the most aggressive, and therefore dominant over smaller individuals (*Hutchings, Bishop & McGregor-Shaw, 1999*). Furthermore, the existence of aggressiveness-related behavioural types has been confirmed in European seabass (*Dicentrarchus labrax*), (*Millot et al., 2014*).

Razorfishes are a ubiquitous group of fish species that inhabit most of the shallow sedimentary habitats of tropical and temperate seas (*Nemtzov & Clark, 1994*; *Battaglia et al., 2010*). Socially, they are characterised by a complex haremic structure formed by a male and several females (4–6 females per male). The male defends his territory and the females that live in it from other males (*Marconato, And & Marin, 1995*; *Cardinale, Colloca & Ardizzone, 1998*; *Shen & Clark, 2016*). However, nothing is known on how behavioural types could be playing a role in the formation of harems and other aspects of the social structure of this marine species. The pearly razorfish (*Xyrichtys novacula*), is the only species of the genus *Xyrichtys* found in the Mediterranean Sea (*Candi et al., 2004*). This species is a small-bodied wrasse that inhabits clear shallow waters with sandy bottoms and sometimes areas covered by the seagrasses *Cymodocea nodosa* and *Zostera marina* (*Castriota*

*et al., 2005*; *Battaglia et al., 2010*). This species is distributed at depths between 0 and 50 m (*Fischer, Schneider & Bauchot, 1987*). It is a diurnal species and it buries itself in the sand to rest and to avoid predators during the night-time (*Alós, Cabanellas-Reboredo & Lowerre-Barbieri, 2012*; *Alós, Martorell-Barceló & Campos-Candela, 2017*). The pearly razorfish has a carnivorous diet based on small benthic invertebrates, which differs depending on the area but basically consists of molluscs and crustaceans and occasionally of echinoderms and polychaetes (*Cardinale, Colloca & Ardizzone, 1997*; *Castriota et al., 2005*; *Battaglia et al., 2010*).

The pearly razorfish is a protogynous, monandric hermaphrodite. It has an appreciable sexual dimorphism showing differences in head shape, length of pelvic fin and colouration patterns, with females presenting a pearly spot on the abdomen (*Cardinale, Colloca & Ardizzone, 1998*; *Candi et al., 2004*). In the pearly razorfish, aggressiveness plays a fundamental role in the establishment and maintenance of harems. Males are highly territorial and defend their harem from other males in contiguous territories; male attacks against females have not been detected (*Shen & Clark, 2016*). The reproductive period of this species takes place during the summer (July to September), and only dominant males participate in spawning (*Marconato, And & Marin, 1995*; *Cardinale, Colloca & Ardizzone, 1998*). Despite the importance of this behavioural trait, the individual repeatability for aggressiveness has never been explored in this species. Therefore, the pearly razorfish offers a unique opportunity to understand how an individual's aggressiveness can determine its position in a complex social system.

The main objective of this study is to quantify for the first-time the aggressiveness in the pearly razorfish and to determine the existence of behavioural types in this species. We determined the repeatability of aggressiveness using metrics obtained under laboratory conditions applying a standardised mirror test. We explored the effects of body size, sex, individual condition and the effect of captivity and isolation in the expression of aggressiveness using generalised linear mixed models and the decomposition of the phenotypic variance into between- and within-individual components.

Our working hypotheses were (i) individual aggressiveness is predictable—aggressiveness-related behavioural types exist, (ii) aggressiveness levels are higher in males and larger individuals (as a response to an increased territorial defence), and (iii) aggressiveness is negatively correlated with the individual condition, as aggressive individuals spend more energy in antagonistic interactions to protect their territory. Our work represents the first study on behavioural types in razorfishes, providing the first step to understand the causes and consequences of individual behaviour in this marine fish group.

## MATERIAL AND METHODS

### Origin of the experimental fish

From April 30th to July 12th 2019, a total of 6 weekly sessions of experimental fishing were carried out in shallow waters (10–15 m deep) of the Marine Protected Area in the Palma Bay, Mallorca (39°27′56″N 02°43′37″E). Twelve fish were caught during each

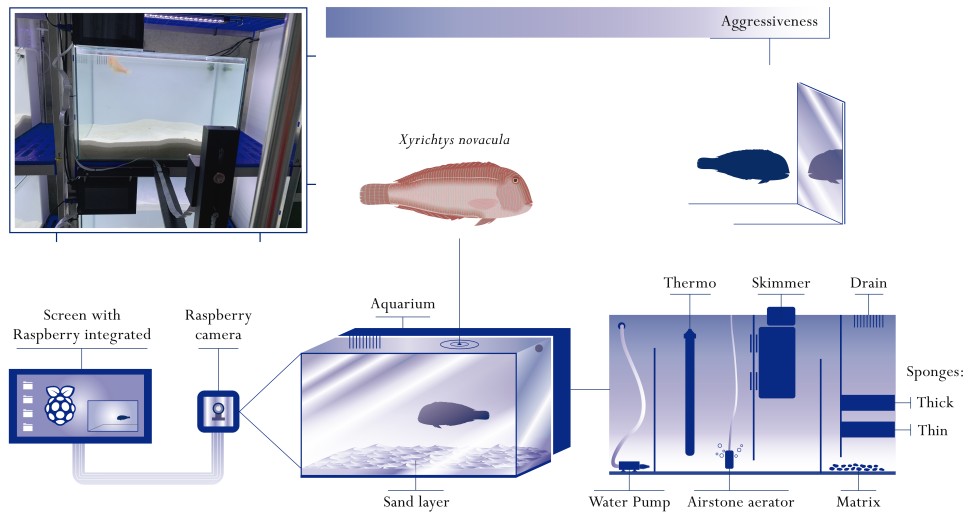

**Figure 1** **Representative diagram of the behavioural arenas.** Behavioural arenas were composed by an aquarium (container where the fish were held captive) connected to the filtering water system through a small sink. The tank contained seawater and sand that served as refuge for the fish. A filtration system was installed behind the tank, consisting of two sponges of different thickness, a biological filtration system and a skimmer responsible for removing the excess of urea. An airstone aerator, a water heater and a water pump were located in the filtration section. The recording system consisted of a Raspberry Pi computer and a Raspberry Pi camera. This system was mounted in front of the aquarium to have a frontal vision during the aggressiveness tests. On the photo, a fish can be seen swimming in an aquarium during the experimental period (diagram by Javier Sanllehi Hansson).

experimental fishing session (this number was limited by the available experimental arenas, see below). Each fishing session was carried out during the morning, from 10:00 to 14:00, using standardised hook-and-line gear and live shrimp as bait (a classical capture method used by local recreational anglers). Fish with deep-hooking (i.e., hooks reaching the internal/vital parts of the body) were not used for this behavioural study to ensure that all individuals presented the least disturbance associated with the capture method. Once 12 individuals were captured (after up to 1 h of sampling), fish were transported in an aerated 80 l container with a constant oxygen flux from the capture site to the Marine Research and Aquaculture Laboratory (LIMIA) in Andratx, Mallorca (39°32′39″N 02°22′42″E). At the LIMIA, fish were individually housed in the experimental behavioural arenas for posterior behavioural testing (Fig. 1). Neither mortalities nor stress (assessed through visual inspection of the body) due to captivity and transport was observed.

## Characteristics of the behavioural arenas

The present study constitutes the first attempt to maintain the pearly razorfish in social isolation. The study of behavioural types in the pearly razorfish constitutes a new challenge because this species requires specific attributes of the experimental arenas to simulate its natural environment. In natural conditions, the pearly razorfish buries in the sand at night to avoid predators (*Alós, Cabanellas-Reboredo & Lowerre-Barbieri, 2012*; *Alós, Martorell-Barceló & Campos-Candela, 2017*), and thus it is necessary to include the appropriate

substrate in the experimental arenas to provide this particular type of shelter. We developed several prototypes of behavioural arenas before carrying out the experimental trials. The optimal behavioural arena was composed by an aquarium of 120 l with closed UV-purified seawater re-circulation that allowed for complete control of light and temperature (see Fig. 1). Each aquarium had its sump station where a protein skimmer, bio-media for biological filtration and aeration maintained the optimal conditions during the week (Fig. 1). Each behavioural arena had the same quantity of sand (20 kg), which generated a ∼5 cm high accumulation of sand at the bottom with a precise granulometry (0.5–1.2 mm) simulating the pearly razorfish habitat, allowing the fish to use the sand as refuge.

The closed water system facilitated the control of environmental variables (such as temperature) and facilitated the maintenance of the aquarium throughout the experimental period. The sand, filters and the aquarium walls were cleaned and the water was changed before a new individual was introduced in the behavioural arena. The temperature of each behavioural arena was controlled by a water heater located in the sump of each aquarium, and it was programmed to remain constant at 21 °C (mean and s.d. of the water temperature for all experiments was 21.1 ± 1.8 °C). We used a light screen EHEIM powerLED + installed at the top of each behavioural arena to reproduce the natural conditions of light/dark cycle, which was automatically controlled with an EHEM LEDcontrol +. We reproduced a natural photoperiod of 12 h of light and 12 h of dark in which the sunrise was set at 07:00 and the sunset at 19:00. In this way, we reproduced the same environmental conditions for all individuals, an essential requirement to study behavioural repeatability (*Réale et al., 2007*).

From all individuals captured, we only considered the ones from which we had at least two days of experimental testing recordings. Due to some initial problems with the video generation and storage of the novel tracking system (see below), we had to discard data from some individuals. For this study, we used the data from 49 wild pearly razorfish (37 females and 12 males) with an average total length (TL) of 15.2 ± 2.4 cm and total weight (TW) of 42.80 ± 20.8 g (Fig. 2). Pearly razorfish were visually sexed as this species shows sexual dimorphism. Significant differences in both TL and TW were observed between sexes as expected for a protogynous hermaphrodite. Each individual was isolated in an experimental arena for one week (three days for acclimation and four days for behavioural testing) and was assigned an individual identifier number (ID). During this period, the individuals were fed 1 g of fresh shrimps per day. We were able to observe that the behaviours recorded in the aquarium were the same as those recorded in the sea. Individuals performed slow swimming, they buried themselves at night, or when scared, and they usually ate every day, suggesting that we succeeded in providing suitable captive habitat for this species.

## Video recording for data collection

During the week in captivity, all individuals were continuously monitored (video recorded in 2-dimensions) using a camera set-up attached in front of each behavioural arena (Fig. 1). The camera set-up was based on a Raspberry Pi system equipped with an 8MP Raspberry Pi Camera Module V2 with a Sony image sensor (Fig. 1). The small camera allowed to record videos at 1080 pixels with a framerate of 30 frames per second. A 128GB USB stick was
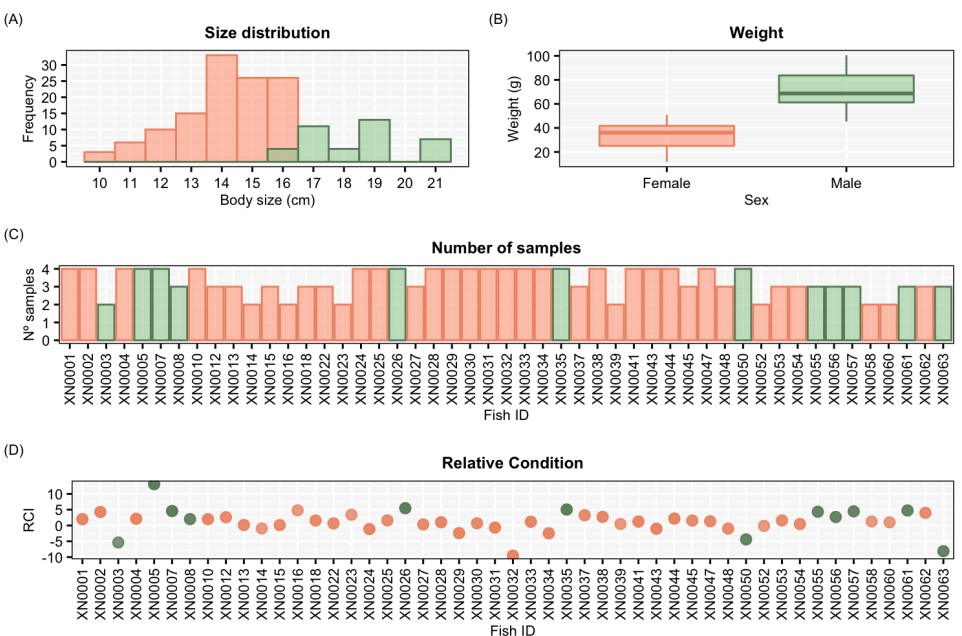

**Figure 2** **Graphical representation of the sample.** (A) Histogram of the distribution of body sizes (in cm) for the individuals considered in this study. (B) Boxplot of the weight differences (in g) between females and males. (C) Bar graph with the number of tests (number of individual samples) performed to each individual. (D) Scatterplot of the relative condition index (RCI) for each individual. Males are represented in green and females in orange.

connected to the Raspberry Pi to store all the generated videos. A screen was attached to the Raspberry Pi where the recordings could be played confirming the correct functioning of the system. This recording system has proven useful for behavioural experiments (*Jolles et al., 2019*). Overall, we provide a new experimental protocol that allows for the monitoring of individual razorfish under laboratory conditions without disturbances (cameras were permanently mounted and recording) that can be used for behavioural quantification in other species.

During the experimental period, approximately 6,900 h of recording were obtained. The advantages of using the continuous recording for behavioural scoring are, (i) raw data is generated and adequately stored for reproducibility analysis, (ii) fish are not unnecessarily disturbed as the camera remains installed during the entire captive period, (iii) it provides with the opportunity to measure behaviour in real-time and, (iv) it provides with the raw data for further automatization of behavioural scoring through machine learning algorithms.

## Standardised aggressiveness test

After the acclimation period (first three days), the individual behavioural assessments using the conventional mirror test started (*Balzarini et al., 2014*; *Way et al., 2015*). Each fish was subject to one test per day for four consecutive days. During testing, a mirror of the same size as the lateral side of the behavioural arena was externally attached to each aquarium

for one hour during the daytime. Three different metrics were visually evaluated from the videos to estimate individual aggressiveness: (i) the number of bites, defined as the number of times the individual bit the mirror, it was considered a bite when fish approached the mirror with their mouth open (*Francis, 1990*; *Budaev, Zworykin & Mochek, 1999*; *Bell & Stamps, 2004*; *Bell, 2005*; *Scotti & Foster, 2007*; *Snekser et al., 2009*), (ii) the number of rams, described as a fast approach with physical contact to the mirror but with the mouth closed (*Balzarini et al., 2014*) and, (iii) the number of charges, described as a fast swim towards the mirror but without direct contact (*Scotti & Foster, 2007*; *Bell, Henderson & Huntingford, 2010*; *Ryan White, 2015*). We were not able to obtain the four tests for all individuals due to problems with the storing of audio-visual material, but for 49 individuals a minimum of two measurements were obtained (see sample size in Fig. 2).

## Data analysis

The degree of consistency in individual behaviour is usually quantified with the repeatability (*R*) score (*Nomakuchi, Park & Bell, 2009*). The *R* score describes the proportion of phenotypic variance that is explained by among-individual differences by decomposing between- and within-individual variance thus defining behavioural types (*Dingemanse & Dochtermann, 2013*). The raw *R* score was computed as the quotient between the between-individual variance ($Vind_0$, the variance across random intercepts of individuals) and the within-individual variance ($Ve_0$, residual variance, the variance associated with measurement error and phenotypic flexibility) for a given behavioural score (*Nakagawa & Schielzeth, 2010*; *Dingemanse & Dochtermann, 2013*). We used a Generalised Linear Mixed Model (GLMM) to decompose variances to estimate *R* and explore contextual effects using a Bayesian approach (*Hadfield, 2010*). To use a single aggressiveness score in the model, we carried out a Principal Component Analysis (PCA) with the three behavioural metrics (number of bites, number of rams and number of charges) at their original scale (counts).We considered the values of the Principal Component 1 (PC1) as aggressiveness score for fitting the GLMM (Fig. 3).

Regarding the contextual variance and the structure of the fixed part of the GLMM, we considered three variables: body size (related to sex), RCI and experimental day. Considering the effects of these variables, we estimated the adjusted-*R* scores —adjusted *R* after controlling for the confounding fixed effects (*Dingemanse & Dochtermann, 2013*). The sequential hermaphroditism of the pearly razorfish inevitably implies that body size and sex are correlated, creating a co-linearity conflict in our GLMM. We selected the fish body size (in cm) as a continuous variable to adjust the GLMM, but the results are discussed as the body size / sexual effect. We computed the RCI, a measure of the internal state of the individual, as the ratio between the observed fresh weight (in g) and the predicted weight from an independently estimated length-weight relationship for the species (*Battaglia et al., 2010*) following the protocol in *Morgan (2004)*. The RCI was preferred because, in contrast to Fulton's condition index, it is independent of body size (*Morgan, 2004*). Finally, the experimental day (an integer number from 1 to 4, assigned depending on the actual experimentation date) was included in the model to control for potential effects of captivity (e.g., acclimation effect). The three variables were scaled and

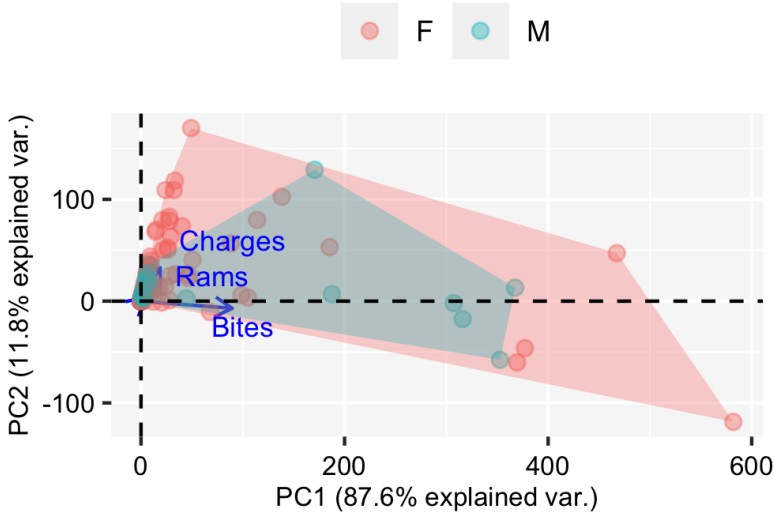

**Figure 3 PCA (Principal component analysis).** Graphical representation of the two components (PC1 and PC2) from the principal component analysis (PCA) and the three correlated aggressiveness metrics (number of bites, rams and charges). Males are represented in green and females in orange.

mean-centred as they represented factors in very different units. The fixed structure of the full model (considering all variables) was reduced using bidirectional elimination (i.e., 'step-by-step' backward reduction, which is a combination of the backward and forward stepwise selection) until the lowest Deviance Information Criteria (DIC) was attained (we assumed a cut-off of two points in the process).

The full GLMM considered a random structure based on a nested variable attributable to the individual nested in experimental weeks (as described in the experimental protocol). The reduction of the model was based on bidirectional elimination, the same strategy used for the fixed structure of the GLMM. The parameters and the Bayesian Credibility Intervals (BCI, 2.5% and 97.5%) of the LMMs were estimated using a Bayesian approach with the default settings from the library MCMCglmm (*Hadfield, 2010*). In all cases, convergence of the chains was attained and checked by plotting the distribution of the residuals. A transformation (natural logarithm) was applied to the cases where the normality of the residuals was not fulfilled. We used the reduced GLMM to compute adjusted-$R$ scores for each fish. The BCI of the adjusted-$R$ scores was interpreted to detect the presence of behavioural types. A likelihood ratio test (LRT) was used to calculate the significance of the adjusted-$R$. According to the LRT, the reduction in the DIC ($\Delta$DIC) provided by the GLMM where $Vind_0$ was constrained to 0 was used to detect significant $Vind_0$. Adjusted-$R$ scores were considered significant when comparing the unconstrained GLMM to the constrained GLMM resulted in DIC reductions larger than 2. To check if there was an effect of TL within the same sex, we separately analysed the data corresponding to the females ($n = 37$) following the same protocol explained above. We could not do this for the males due to the small sample size ($n = 12$).

## Ethical notes

This study was positively evaluated by the Ethical Committee for Animal Experimentation of the University of the Balearic Islands (references for the protocols CEEA 97/0718 and CEEA 107/01/19), and authorised by the Animal Research Ethical Committee of the Conselleria d'Agricultura, Pesca i Alimentació and the Direcció General de Pesca i Medi Marí of the Government of the Balearic Islands (reference for the authorisations ES070050000502 and 2019/20/AEXP).

# RESULTS

## Sample size and general results

Aggressiveness was successfully scored (at least twice) for all 49 individuals. In all tests, the three metrics for aggressiveness (number of bites, number of rams and number of charges) were successfully quantified. The first two components of the PCA applied to the three aggressiveness' metrics are shown in Fig. 3. The PC1 explained 87.6% of the total variance, showing a correlation betwenn the three variables (Fig. 3). We therefore selected the PC1 scores as a unique measure for aggressiveness. The aggressiveness score defined by the PC1 was normalised to a natural number by multiplying it by a factor of $-1$, so the higher scores described the more aggressive individuals with larger number of bites, number of rams and number of charges (Fig. 3), and the lower scores described lower levels of antagonistic behaviours in front of the mirror (Fig. 3).

## Effect of contextual variables in the aggressiveness score

The estimates and confidence intervals (CI) for the effects of the contextual variance on the aggressiveness score of the pearly razorfish are shown in Table 1. The GLMM suggests that neither the body size nor the sex have a significant effect on the aggressiveness of the pearly razorfish (Table 1 and Fig. 4). Results were consistent in the female-only data set being the scores of small and large females similar (Table 2 and Fig. 5). The individuals presented a mean $\pm$ s.d. RCI of $1.28 \pm 3.65$ (range of $-9.53$ to $13.07$). The results regarding the RCI were also non-significant in both GLMMs, suggesting that the condition of the fish was not related to its aggressiveness score (Table 1). Finally, the effect of the experimental day was also non-significant in both models, rejecting any effect of the days under laboratory conditions on aggressiveness levels (Table 1). In both fitted GLMMs (complete and female-only data set), only the ID of the individual (between-individual variance) was retained in the random structure of the model according to the DIC optimisation, rejecting an among-week effect.

## Adjusted repeatability and bayesian credibility intervals

The aggressiveness scores for the complete data set averaged $33.38 \pm 91.15$ (Fig. 4), and their adjusted-$R$ score was 0.57 with a BCI of [0.44–0.60] (Table 1). For the female-only data set, the average aggressiveness was $28.98 \pm 85.38$ (Fig. 5), and their adjusted-$R$ score 0.36 with a BCI of [0.27–0.48] (Table 2). The fit and the DIC of the constrained (DICc) GLMM suggested that this adjusted-$R$ score could be considered significant in both cases, evidencing the existence of aggressive and non-aggressive behavioural types

**Table 1** **Results of the GLMM for the complete data set (including males and females).** Environmental covariates with their lower and upper BCI (l-BCI and u-BCI, respectively) are shown. The model was fitted by three environmental variables: experimental day, body size (related with sex: the males were bigger than the females); and the relative condition index (RCI). None of these variables have an effect on aggressiveness. The table also shows the adjusted-R score, statistically significant as evidenced by the values of DIC and DICc, the between- (Vind0) and within-individuals (Ve0) variances and $p$-values.

| | Mean | l-BCI | u-BCI | p-value |
|---|---|---|---|---|
| Intercept | 1.68 | 1.27 | 2.12 | <0.001* |
| Experimental day | 0.17 | −0.03 | 0.37 | 0.09 |
| Body Size | 0.12 | −0.32 | 0.59 | 0.59 |
| RCI | −0.08 | −0.50 | 0.39 | 0.73 |
| $V_{ind0}$ | 1.90 | 1.08 | 3.02 | |
| $V_{e0}$ | 1.54 | 1.18 | 1.95 | |
| Adjusted-$R$ | 0.57 | 0.44 | 0.60 | |

DIC = 558.22 (DICc = 641.32)

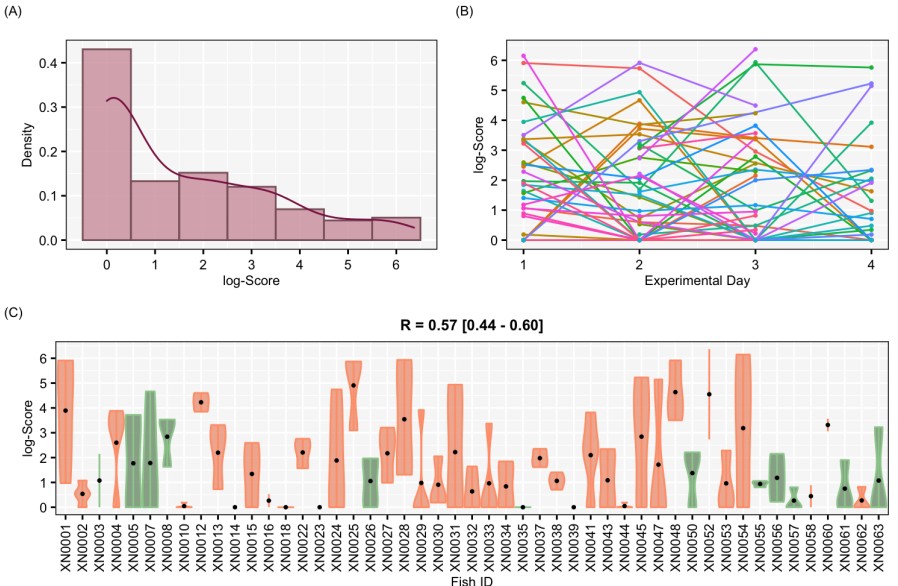

**Figure 4** **Results for the complete data set** Results of aggressiveness scores obtained from the complete data set (including males and females). (A) Density population plot. (B) Individual's aggressiveness score per each experimental day, where each colour represents an individual. (C) Violin plot representing the individual's variance of aggressiveness scores, where each dot represents the mean aggressiveness score for each individual. Females are represented in orange and males in green.

(Tables 1 and 2). The model shows that the $R$ was significant: DIC = 558.22 and DICc = 641.32 (for the complete data set. Table 1) and DIC = 481.33 and DICc = 446.62 (for the female-only data set, Table 2). This was reflected by a large phenotypic variation ranging from individuals who bit on average 350 times per test (for example, XN0035, Fig. 4) and individuals that did not interact with the mirror (for example, XN0047, Fig. 4), independently of their sex and size.

**Table 2  Results of the GLMM for the female-only data set.** Environmental covariates with their lower and upper BCI (l-BCI and u-BCI, respectively) are shown. The model was fitted by three environmental variables: experimental day, body size and the relative condition index (RCI). None of these variables have an effect on aggressiveness behaviour as in the complete data set. The table also shows the adjusted-R score, statistically significant as evidenced by the values of DIC and DICc, the between- (Vind0) and within-individuals (Ve0) variances and *p*-values.

| | Mean | l-BCI | u-BCI | *p*-value |
|---|---|---|---|---|
| Intercept | 1.65 | 1.20 | 2.13 | <0.001* |
| Experimental day | 0.14 | -0.11 | 0.40 | 0.31 |
| Body size | 0.10 | -0.37 | 0.56 | 0.68 |
| RCI | 0.02 | -0.39 | 0.45 | 0.90 |
| $V_{ind0}$ | 1.42 | 0.45 | 2.66 | |
| $V_{e0}$ | 1.75 | 1.37 | 2.61 | |
| Adjusted-*R* | 0.36 | 0.27 | 0.48 | |
| DIC = 481.33 (DICc = 446.62) | | | | |

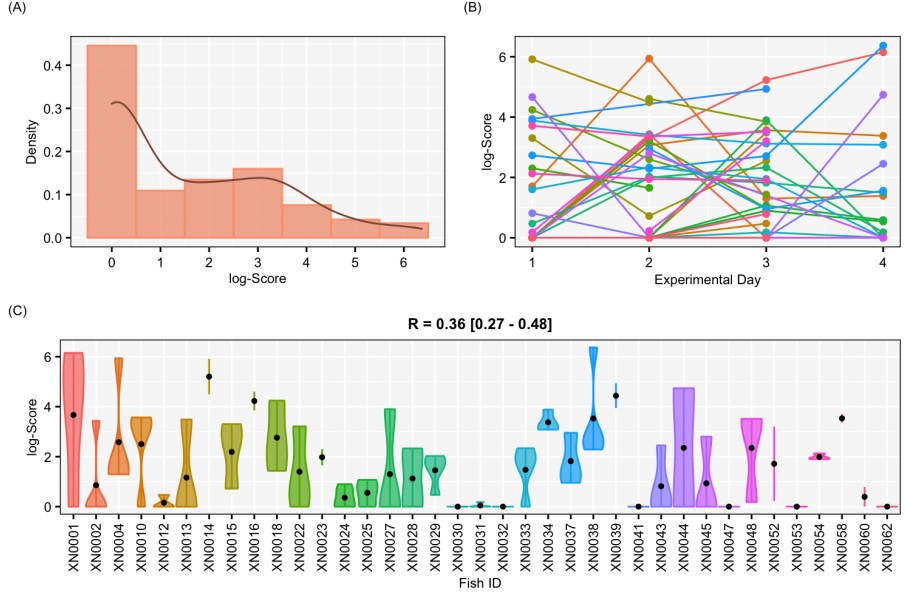

**Figure 5  Results for the female-only data set.** Results of aggressiveness scores obtained from the female-only data set. (A) Density population plot. (B) Individual's aggressiveness score per each experimental day, where each colour represents an individual. (C) Violin plot representing the individual's variance of aggressiveness scores, where each dot represents the individual's mean.

## DISCUSSION

This work is the first behavioural study considering captive razorfishes. Aggressiveness was tested on 49 pearly razorfish and our results showed a *R* score for aggressiveness of 0.54 [0.48–0.61]. The *R* score was statistically significant, demonstrating that consistent differences among individuals exist and corroborating the existence of aggressiveness-related behavioural types in the pearly razorfish. Most importantly, assuming a *R* score

of 0.54 (average of our values), 49 individuals (with an average of three experimental trials per individual) provided an accuracy (root mean square error) of ~0.1 for estimating phenotypic behavioural variation, and a power greater than ~0.8 for estimating behavioural syndromes following *Dingemanse & Dochtermann (2013)*. The variables considered in this study (body size and sex, RCI and experimental day) did not affect the aggressiveness score of the pearly razorfish. Therefore, we rejected our initial hypotheses that males were more aggressive than females and that the most aggressive individuals were those with the lowest RCI. These results suggest that aggressiveness is intrinsic to the individual's life history, and that aggressiveness may be involved in different aspects of the ecology and biology of the pearly razorfish. Moreover, the fast acclimatisation, the low-stress levels observed, and the fact that the individuals responded to the mirror, confirms that our system is adequate to measure aggressiveness in pearly razorfish, making this species a successful candidate for behavioural ecology studies in marine fish.

Our results suggest that the mirror test is an excellent methodology to measure aggressiveness in the pearly razorfish, providing real measures of agonistic interactions in front of a conspecific individual. The mirror test has been widely used in fish, with excellent results, which show the optimisation of this test to measure aggressive behaviours (*Francis, 1990*; *Grantner & Taborsky, 1998*; *Rowland, 1999*; *Budaev, Zworykin & Mochek, 1999*; *Vøllestad & Quinn, 2003*; *Pham et al., 2012*). Since fish, like many other taxa, do not recognise their reflection in the mirror (*Gallup, 1968*)—at least if they are exposed to such stimulus for short periods of time as when performing a behavioural test (*Kohda et al., 2019*)—the mirror test entails reinforced aggressiveness given that it provides a continuous stimulation by an opponent (*Gallup, 1968*; *Rowland, 1999*). Furthermore, the mirror test is also used in territoriality studies to determine which male is in a better position to feed and reproduce. In our species, behaviours such as territoriality and dominance of some males over others have been described (*Scotti & Foster, 2007*; *Snekser et al., 2009*; *McGhee & Travis, 2010*; *Blowes, Pratchett & Connolly, 2013*). Overall, the mirror test has some advantages compared to other methodologies such as the opponent test (*Rhoad, Kalat & Klopfer, 1975*). The number of experimental individuals used is reduced to the focal individuals. This is particularly relevant in our study species because individuals bury in the sand when they feel threatened, thus making the capture process within the tank challenging. Placing the ''opponent'' and the focal individual side by side during the trials would significantly increase the individuals' stress and anxiety, affecting the results obtained during the behavioural testing. Moreover, the opponent test requires an accurate selection of opponent fish because size, sex and behaviour may influence the focal individual's response (*Earley, Hsu & Wolf, 2000*; *Wilson et al., 2011*). If different opponents are used, it becomes challenging to compare the results. Obtaining aggressiveness scores for the focal individuals regardless of the opponent's characteristics would require a vast sample size to repeat the experiment with multitude of different opponents.

From assessing aggressiveness in our population, we have obtained a relatively high $R$ score (0.54) compared with other studies. The average $R$ across 98 species is 0.37 (*Bell, Hankison & Laskowski, 2009*). Other studies, however, have reported higher $R$ scores. In the blue catfish (*Luciana godei*), an $R$ score of 0.72 was obtained for aggressive behaviours

(*McGhee & Travis, 2010*). In the zebrafish, specific *R* scores were recorded for different metrics of aggressiveness: an *R* score of 0.39 was obtained for overall aggressive behaviours (*Roy & Bhat, 2018*), an *R* score of 0.56 for the number of bites (*Way et al., 2015*), and a lack of repeatability was reported for mirror tests experiments in zebrafish (*Sbragaglia et al., 2019*). These differences suggest that the *R* score is sensitive to the selected aggressiveness metric. In our study, a PCA was performed to obtain an aggressiveness score (PC1) that was the result of reducing the dimensions of our measured variables. The PC1 explained 87.9% of the variability of the data set, and it was mostly related to the number of bites (Fig. 3), which seems to be the most relevant metric to quantify aggressiveness at the individual level according to previous work (*Way et al., 2015*) and our findings. Our results demonstrated that aggressiveness is a constant behaviour maintained within individuals and overtime in the pearly razorfish (Table 1 and Fig. 4 for the complete data set; Table 2 and Fig. 5 for female-only data set).

Differences in aggressiveness were not related to body size. Since the pearly razorfish is a sequential hermaphrodite, size and sex are correlated, and thus the lack of effect should be jointly interpreted. Other studies revealed a positive relationship between aggressiveness and size in different species such as the lion-headed cichlid, (*Steatocranus casuarius*), (*Budaev, Zworykin & Mochek, 1999*); the green swordtails, (*Wilson et al., 2011*) and the Midas cichlid, (*Francis, 1990*). In general, it has been found that aggressiveness plays an important role in social structure, where the most aggressive males have a higher position in the mating hierarchy (*McGhee & Travis, 2010*). This could be the case of the pearly razorfish, a species that lives in harems formed by a male and several females (*Espino et al., 2015*) and where there is intra-sexual but not inter-sexual competition. Male and female territories usually overlap, suggesting that the shared space is used for reproduction (*Shen & Clark, 2016*). Likely, male aggression is probably related to their position in the social structure and their resource holding potential (RHP), defined as the ability to win a contest and maintain earned resources (*Parker, 1974*; *Hurd, 2006*). RHP depends on the size of the opponents, and the motivation, physiological state and behaviour of the focal individual (*Parker, 1974*; *Nijman & Heuts, 2000*). Thus, aggressiveness on pearly razorfish might be playing an essential role in the structure of the population.

Our initial hypothesis that the RCI was negatively correlated with aggression was also rejected. Individual behavioural responses may modulate internal energetic fluxes. Since aggressive behaviours are energetically very demanding (*Katano & Iguchi, 1996*), highly aggressive individuals may have a poor condition because they invest a lot of time and energy in antagonistic behaviours (*Grantner & Taborsky, 1998*). Most studies agree that aggressive behaviours are energetically costly, and different internal growth rates, maturity stages and metabolic rates are negatively related to aggressiveness (*Cutts, 1998*; *Vøllestad & Quinn, 2003*). In the present study, we did not find a relationship between the RCI and the aggressiveness score. This could be since metabolic levels may vary between seasons due to food availability, while behaviour remains more constant over time (*Vøllestad & Quinn, 2003*; *Bouwhuis et al., 2014*). We also considered the effect of the experimental day in our GLMM to control for potential effects of captivity. Visually, individuals displayed normal behaviour after the three days of acclimation. In the same way, fishes could have habituated

to the mirror during the experiments. Studies have shown that continuous exposure to stimuli over a series of consecutive days can decrease the level of behavioural reactivity due to habituation. Long acclimatisation could reduce fish activity levels (*O'Neill et al., 2018*), decrease exploration rates and increase individual's boldness over time (*Adriaenssens & Johnsson, 2011*). The effects of acclimatisation have also been reported on aggressiveness; in fact, the number of aggressive interactions can decrease significantly across trials (*Roy & Bhat, 2018*). In the pearly razorfish, however, the correlation between experimental day and aggressiveness score was not statistically significant, rejecting any acclimation or habituation effect.

Our results suggest that aggressiveness is a significant individual intrinsic factor, and we can derive some potential physiological and ecological consequences from the existence of aggressiveness behavioural types in the pearly razorfish. Aggressiveness has been closely related to growth and metabolic rates (*Cutts, 1998*; *Vøllestad & Quinn, 2003*), reproduction (*Hirschenhauser et al., 2008*; *Snekser et al., 2009*; *McGhee & Travis, 2010*) and dominance in social structures (*Grantner & Taborsky, 1998*; *Earley, Hsu & Wolf, 2000*; *May & Mercier, 2007*; *Pham et al., 2012*; *Blowes, Pratchett & Connolly, 2013*). Therefore, aggressiveness could be linked to the life history of the pearly razorfish affecting the individual fitness and its position within its social group. Instead of expecting a difference in aggressiveness related to sex, size or condition, we could expect differences in aggressiveness linked to the social structure of this species. We could also expect aggressiveness to be related to the formation and maintenance of individual territories. Following this idea, the most aggressive males would be the ones obtaining the best feeding patches and the highest quality mates. Similarly, females could compete with each other for the best males. These hypotheses deserve more attention and require the combination of measurements of aggressiveness scores in the laboratory alongside with the measurement of territory sizes in the field.

The relationship between dominance and aggressiveness extends across taxa (*May & Mercier, 2007*; *Snekser et al., 2009*). For instance, in salmonid species, the most aggressive individuals have better competitive capabilities, making them dominant and monopolize high-quality territories (*Huntingford, 1998*). Aggressive individuals defend high-quality territories since the benefit obtained from this territory is greater than the cost of defending it (*Bacon, Ball & Blackwell, 1991*; *Stamps & Krishnan, 1999*). Aggressiveness plays a crucial role in obtaining and maintaining territories. Similarly, the attack of an intruder to obtain a territory will cease if the benefit obtained is not high enough (*Stamps & Buechner, 1985*; *Stamps & Krishnan, 1999*). Animals' ability to achieve and defend their territories depends mostly on their behaviour (*Stamps, 1994*; *Dingemanse & Réale, 2005*). Therefore, our work demonstrating the existence of aggressive behavioural types provides a novel perspective on understanding the territorial societies of the pearly razorfish.

Furthermore, it is necessary to understand how disturbances affect fish behaviour and populations (*Biro, Post & Booth, 2007*; *Biro, Beckmann & Stamps, 2010*). The study of behavioural types is essential to understand ecological patterns and population dynamics (*Conrad et al., 2011*; *Sih et al., 2012*). Intraspecific variability in fish behaviour is closely related to the resilience of populations to environmental disturbances (e.g., climate

change) (*Wolf et al., 2007*). It has been shown that anthropic disturbances such as fishing have important effects on behavioural types (*Biro, Post & Booth, 2007*; *Biro, Beckmann & Stamps, 2010*). Fishing is one of the biggest drivers of evolution in fish species (*Sullivan, Bird & Perry, 2017*). Fish vulnerability to fishing is a complex phenomenon in which many processes are involved, and in which behaviour plays a significant role (*Arlinghaus et al., 2017*). It has been demonstrated that fishers select for certain behaviours (activity, awakening time) in wild pearly razorfish populations (*Alós et al., 2016*; *Martorell-Barceló, Campos-Candela & Alós, 2018*). In other species, fishing selects against bold and aggressive individuals (*Arlinghaus et al., 2017*). This could also be the case in the pearly razorfish populations. Since the most aggressive individuals tend to take more risks in dangerous situations, their removal would likely reduce the likelihood of them being killed by natural predators, (*Sih et al., 2012*; *Arlinghaus et al., 2017*). In this way, the intraspecific variability of aggressiveness in the pearly razorfish can be reduced in exploited populations, affecting a multitude of ecosystem processes with significant ecological and economic consequences. For all the reasons mentioned above, this research deserves further attention and the development of experimental protocols involving laboratory scoring and wild assessment of behaviour in real exploited populations.

In this study, we provide, for the first time, with a behavioural type scoring for the pearly razorfish. However, our research has some limitations. First, we obtained the experimental fish via experimental fishing, and we provide evidence that fishing selects a type of behaviour, skewing the sampled population. Despite this, our individuals were obtained from a population located in a Marine Protected Area where fishing is not allowed, and therefore the entire repertoire of behaviours should be vulnerable. Furthermore, the sampling equipment used was based on natural baits. Generally, the link between hook and line vulnerability and aggressiveness has been found in fisheries using artificial lures (*Lennox et al., 2017*). Therefore, in our opinion, our sample was representative of the studied population. Second, we tested a small sample of males ($N = 12$). It is possible that increasing male's sample size would translate into significant differences in aggressiveness associated with size. In the present study, we only considered aggressiveness. However, in future works, it would be interesting to study behavioural syndromes between aggressiveness and other behavioural types, including chronotypes.

## CONCLUSIONS

Our work shows, for the first time, the presence of behavioural types in the pearly razorfish, provided by a significant and high repeatability score. Body size (correlated with sex), RCI and the experimental day did not have a significant effect on the aggressive behaviours of the pearly razorfish. Aquariums with a closed water system and the mirror test have proven successful as a methodology for studying aggressive behaviour in the pearly razorfish. The results obtained from the mirror test revealed consistent differences in aggressiveness levels among individuals. This is the first behavioural study on the pearly razorfish under controlled laboratory conditions, its fast acclimatisation and low-stress levels observed make this species a successful candidate for behavioural studies of marine fish. This study

also provides novel information on fish aggressiveness. We believe this is the first step for a better understanding of the pearly razorfish personality and its implications in ecological and evolutionary processes.

## ACKNOWLEDGEMENTS

We thank the staff of the Marine Research and Aquaculture Laboratory of Andratx (LIMIA) for the help received and for letting us use their facilities for this study. This work is a contribution of the joint research unit IMEDEA –LIMIA.

### Funding

This work was carried out as part of the research project Cronofish (AAEE 101/2017) funded by Balearic Islands Government. In addition, this project also received financing from the CLOCKS project from the Spanish Government (PID2019-104940GA-I00). Martina Martorell-Barceló was supported by an FPI predoctoral fellowship (ref. FPI/2167/2018) from the Balearic Islands Government General Direction of Innovation and Research. Josep Alós was supported by a Ramon y Cajal Grant (grant no. RYC2018-024488-I) and the intramural research project JSATS (grant no. PIE 202030E002) funded by the Spanish Ministry of Science and Innovation, and the Spanish National Research Council. The funders had no role in study design, data collection and analysis, decision to publish, or preparation of the manuscript.

### Grant Disclosures

The following grant information was disclosed by the authors:
Balearic Islands Government: AAEE 101/2017.
CLOCKS project: PID2019-104940GA-I00.
FPI predoctoral fellowship: FPI/2167/2018.
Ramon y Cajal Grant: RYC2018-024488-I.
JSATS: 202030E002.

### Competing Interests

The authors declare there are no competing interests.

### Author Contributions

- Martina Martorell Barceló conceived and designed the experiments, performed the experiments, analyzed the data, prepared figures and/or tables, authored or reviewed drafts of the paper, and approved the final draft.
- Júlia Mulet performed the experiments, analyzed the data, prepared figures and/or tables, authored or reviewed drafts of the paper, and approved the final draft.
- Javier Sanllehi, Marco Signaroli, Margarida Barcelo-Serra and Eneko Aspillaga performed the experiments, authored or reviewed drafts of the paper, and approved the final draft.
- Arancha Lana performed the experiments, authored or reviewed drafts of the paper, set up the entire Raspberry system, and approved the final draft.

- Josep Alós conceived and designed the experiments, analyzed the data, prepared figures and/or tables, authored or reviewed drafts of the paper, and approved the final draft.

## Animal Ethics

The following information was supplied relating to ethical approvals (i.e., approving body and any reference numbers):

This study was positively evaluated by the Ethical Committee for Animal Experimentation of the University of the Balearic Islands (CEEA 97/0718 and CEEA 107/01/19).

## Field Study Permissions

The following information was supplied relating to field study approvals (i.e., approving body and any reference numbers):

This study was authorized by the Conselleria d'Agricultura, Pesca i Alimentació and the Direcció General de Pesca i Medi Marí of the Government of Balearic Islands (Reference for the authorizations ES070050000502 and 2019/20/AEXP).

## Data Availability

Data for this study is available at the DIGITAL.CSIC repository:

Martorell-Barceló, Martina; Mulet, Júlia; Sanllehi, Javier; Signaroli, Marco; Lana, Arantxa; Barceló-Serra, Margarida; Aspillaga, Eneko; Alós, Josep; 2021; Aggressiveness-related behavioural types in the pearly razorfish [dataset]; DIGITAL.CSIC; http://dx.doi.org/10.20350/digitalCSIC/13682.

The raw data is available in the Supplemental Files.

## Supplemental Information

Supplemental information for this article can be found online at http://dx.doi.org/10.7717/peerj.10731#supplemental-information.

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
