# Peer review of "Aggressiveness-related behavioural types in the pearly razorfish"

_PeerJ, doi:10.7717/peerj.10731_

## Round 0.1 · original submission · Minor Revisions

I read your paper with interest as individual variation is a rapidly growing topic of central importance to many other issues of animal cognition, behavior and welfare. I was fortunate to receive two very helpful reviews from experts, one of whom recommends only minor revision. The other had more substantive suggestions for reframing your introduction among other things. I agree with their comments and have a few of my own.

You mention catching 12 fish across six days – what happened to 13 of the fish that were not tested, giving you 49 study animals?
You have a relatively small sample with a very small number of observations for this type of study. Can you address how typical this is for studies of this kind?

You should cite the paper by Kohda et al. on mirror self-recognition in a fish, along with some of the related commentaries around line 86.
Is it possible that size is not a factor when using a mirror test b/c the size of the ostensive opponent is matched to the subject? In the discussion, please be clear when other tests of aggression have used the same test or if they used a different test.

Is the variance within subjects less than the variance between? That is not clear from your data with the graphs seeming to show quite a bit of variability across test days (even if that was not significant in the model).

Line 53, replace “poorly” with “been.”
Line 57 delete extra )
Line 63, move the “being” to after “test”
Line 85, change “able” to “capable.”
Line 115, missing “of” after “existence.”
Line 180 “trough” should be ”through.”
Line 186 delete “of”
Line 402, place ‘ after individuals and place “an” before effect.

Reviewer 1 ·

Basic reporting

- Basic reporting is generaly fine.

- Some article should be added. In particular articles realted to ecological aspects of aggresiveness in similar marine species and studies that criticize the mirror test.

- Hypotheses should be better contextualized

Experimental design

- Research within the scope of journal

- Some work to be done on contextualizing research hypotheses

- Information is sufficient to replicate the study

Validity of the findings

- The finding are valid and correctly exposed to the reader.

- A specific paragraph should be added to address limitations and possible biases of sampling methods.

Additional comments

I read with great interest the paper entitled “Aggressiveness-related behavioural types in the pearly razorfish”. I think it is an interesting study that deals with individual personality of a very interesting model system. I think the model system presented in this paper and the methodological approach is a very promising research tool and I hope my comments will help the authors to improve the quality of the manuscript.

The topic in interesting and timely, however I have several major concerns that are listed below together with some minor comments.

1) I think the title needs some rethinking. The paper investigates the behavior of fish in front of a mirror and the results do not demonstrate that this is “aggressiveness-related”. It is an open discussion in the literature. The authors reported some papers about it (e.g., Balzarini et al. 2014), but I think they cannot demonstrate if the behaviours presented here are aggressiveness-related or not. This aspect should be made clear.

2) I would avoid terms such as “for the first time” or “novel”. I think it is not necessary.

3) The introduction ends with the statement of the working hypotheses. However, I think a that the introduction needs more work to better prepare the reader for the working hypotheses. The great part of the information is already in the introduction, but needs a better logical structure. I tried to make some suggestion in the following points.

4) The first paragraph of the introduction introduces the topic of personality trait, but I think that “Behavioural types and syndromes are widespread across taxa (Réale et al., 2007), although its causes and consequences in marine fish species have scarcely poorly considered (Hutchings, Bishop McGregor-Shaw, 1999; de Jong et al., 2009; Millot et al., 2014)” is not a good justification for the study. What I mean is that the first paragraph of the introduction should end with the overall importance of the study, but I think the fact that something is scarcely considered in marine fishes is (by itself) not a major justification to investigate it. I think this paragraph should be refocused by removing all technical information related to “R”, which suits more in the methods.

5) Conrad et al 2011 is used as main reference for personality traits, however I think Réale et al 2010 should also considered here. Moreover, several sentences need a proper reference such as lines 62-63. Please check the rest of the manuscript for similar situations.

6) There is an open discussion in the literature regarding the mirror test. I think the author did a good but not complete job in addressing it. Please check this paper:
Li, C.-Y., C. Curtis, and R. L. Earley. 2018. Nonreversing mirrors elicit
behaviour that more accurately predicts performance against live
opponents. Animal Behaviour 137:95–105.
Did you use a normal or non-reversing mirror in your experiments?

7) Regarding working hypothesis (ii) I think this is something well established, but still worth testing. However, the authors should do a better job in introducing previous studies and also the concept of resource holding potential, which is totally missing in the paper.

8) Regarding working hypothesis (iii) I think also in this case a better introduction into the logic of the hypothesis is needed earlier in the text.

9) I have an overall major concern about the sampling method. Did you think about a possible sampling bias associated to the fishing technique? This aspect should be addressed in the paper. You explicitly say that fishing selects specific behaviours at the end of the discussion. What is the impact on such bias on your interpretations?

10) The PCA approach is interesting, but why did you decide to do that? I think it would be more interesting to run one model for each behavior and discuss the results independently. This could be more important at the moment to discuss possible problems related to the mirror test.

11) You have working es so I would not suggest to write the second paragraph on the discussion on a methodological aspect (lines 367-382).

12) I think a “limitations” paragraph is needed in the discussion

13) I would suggest the following references:

- https://onlinelibrary.wiley.com/doi/abs/10.1111/faf.12176
Instead of Arlinghaus et al. 2016

- https://afspubs.onlinelibrary.wiley.com/doi/abs/10.1002/tafs.10160
I think this study regarding a lack of repeatability for the mirror test in zebrafish is worth to be cited. Moreover, some aspect of the selection could be related to the concepts in Arlinghaus et al. 2017.

- https://www.sciencedirect.com/science/article/abs/pii/S0003347217300817
Sociability in your mirror test could be considered or at least mentioned

- https://jeb.biologists.org/content/220/24/4624.short
I think an interesting topic related to the razorfish could be related to the effects of aggressiveness and daily activity rhythms (I have seen you already publish a paper about daily activity rhythms in razorfish). I think this paper could be interesting (despite it deals with lobsters) because it is about a burrowing species such as the razorfish, which forms dominance hierarchy with the same ecological functions as in razorfish. Maybe is worth to be mentioned in the discussion instead of the paragraph related to genetic mechanisms, which comes out of the blue.

MINOR COMMENTS
- I think you could define if and what kind of swimming you recorded in from of the mirror (e.g., parallel swimming”). This aspect is tackled in one of the paper you cited (Balzarini et al. 2014) and I think could be interesting to better support whether the mirror test you used here.
- What does (4-6) represents in line 119?
- What does 1080p/30 in line 225 mean?
- I think it could be worthy to present supplementary videos regarding the behaviours characterized in the mirror test.
- Lines 311-312 (is that useful?). I would rather present range of values.
- Lines 384-385. This is not true. You did not validate the results
- Lines 402. Also isolation in an aquarium increase stress and anxiety in fish.
- Lines 499-508. It seems to come out of the blue

Reviewer 2 ·

Basic reporting

The English language needs some improvement – there are a few typos but also some grammatical errors. I provide some specific examples in detail below.
Figures 4c and 5c are missing the Y-axis label.
The Discussion section could be better structured and more focused on the original research questions.

Experimental design

There are well defined and relevant research questions and the experimental design is adequate to answer them. The methodology, including data analysis is well described.

Validity of the findings

Results are statistically sound, and their interpretation is meaningful. Some directions for future studies are provided and discussed.

Line 20 – consider replacing “in” before pearly razorfish with “of”
Keywords – consider replacing keywords already present in the title
Line 53 – Please rephrase
Line 97 – replace “big” with “larger”
Line 98 – replace “small” with “smaller”
Line 103 – Delete “the leave”
Line 185 – 187 – Rephrase
Line 191 – Replace “trails” with “trials”
Line 191 – How can you say that it was the optimal arena? How did you measure their performance? This 120L arena was at most the best of the tested arenas but this does not necessarily mean it is the optimal. More information on this could be added in the supplements
Line 197-198 – “…pearly razorfish habitat and to allowing the fish to use the sand as refuge.” suggest changing to “…pearly razorfish habitat, allowing the fish to use the sand as refuge.”

Line 202 – “…water of the aquariums was changed…”

Line 212 – “…with an average total length (TL) of 15.2 ± 2.4 cm and…”

Line 215 – Please present the results for the statistical tests

Line 238 – How was the time of the test defined? Was it random? Did each fish had the test at the same time every day?

Line 263 – Why not consider time of day as well? (See previous comment)


Line 296 – Replace “dan” with “than”

Line 390 – Consider replacing “since” with “given that”

Line 394/396 – Not very clear, please rephrase

Line 491/492 – “For example…” – IMO this sentence could be removed since it does not add much relevant information to what is stated before.

Line 515 – Please provide references

Additional comments

Besides the general comments on Basic Reporting, Experimental design and Validity of Findings I have some specific comments that I would like the authors to address

---

## Round 0.2 · Minor Revisions

Thank you for being attentive to the last round of reviews. You will see that one reviewer re-reviewed your MS and has some additional minor comments. Most critically, they suggest modifying the title of your MS. I think you have made a case for why you can use your results to speak about aggressiveness, and I am probably more convinced than the reviewer is, but it does not hurt to be clear about the method used to determine this in the title. I also have some minor comments that I would like you to attend to before the paper can be formally accepted.
PeerJ does not provide copyediting and there are numerous grammatical and typographical errors. On the first two pages alone, I have identified the following. Please edit carefully before resubmitting.

I don’t think that Raspberry Pi or generalized mixed models should be key words for your research

Line 20. Through instead of trough.
Line 39, needs to be a comma at the end.
Lines 41-44 are awkward.
Please insert commas after clauses such as after “fish species” on line 44. Check carefully throughout.
Line 50 is missing a “to” after “due.”
Line 58 “consists of” not “in.”
There should be references at the end of the sentence on line 61.
Line 62, place a ; before “however” and a , after. Check throughout.
Line 73, “unknown”
I appreciated that you provided the fish with substrate to bury in to reduce their stress.
Line 220, please write “discard data from..” rather than “discard some individuals.”
Line 240 “has proven useful”
Line 255, “on” instead of “to each fish.”
Line 352 change “with” to “to.”
Line 506 Animals needs an ‘ at the end.
Line 508, “Undertesting?”
On line 522, you write that “fishing selects for bold and aggressive individuals,” which is confusing because fishing, as a selection pressure, is then selecting for less bold and aggressive individuals, correct? Please clarify the wording here.
Line 538, permitted is misspelled.
Line 557, delete “with”
Line 558, “understanding” not “understating”
Caption for Table 2: delete “de”

Reviewer 1 ·

Basic reporting

Basic reporting is fine and readability is good.

Experimental design

Well explained and with all the major elements to replicate it.

Validity of the findings

Finding are robust, but sometimes overstated.

Additional comments

I think the quality and clarity of the paper strongly increased. However, I still have some concern regarding the way in which the mirror test is presented and discussed.

I am not sure that you can claim that what you measure is aggressive-related. The only way would be to see if exist a correlation between the mirror test and the rank in dyadic encounters or in a dominance hierarchy. I think it is still a valid and appropriate test to assess repeatability of behavior, but I think “aggressive-related” is not the best way to refer to it in the title. Maybe you can simply refer to the mirror test. I am not questioning your results, but I am simply not convinced that the scientific discussion on the validity of this test is over. Therefore, the title of the paper is referring to something that you are not demonstrating with your results. I would strongly suggest to change the title to something like: “The mirror test suggests the existence of aggressiveness-related behavioural types in the pearly razorfish”. In this way you still have “aggression” in the title, but you clearly say how you measured it.

I think you cannot say what you say in lines 625-629. You did not measure stress and you did not validate the mirror test with other approaches to be sure that you are measuring aggressiveness (the same for the rest of the paper). I think more caution is needed in the way in which the results are discussed. They are valid results and they open a variety of opportunity for future studies, but I think some sentences as the once highlighted above are overstated.

Regarding the use of the PCA I suggest to refer to this recent guide where there is a specific section on the use of PCA in statistical analysis related to animal personality (https://onlinelibrary.wiley.com/doi/full/10.1111/eth.13082).

Some minor comments:
- In lines 252 I suppose you are referring only to fish. There is a lot of literature about lobsters that you are not citing. So, I would specify you are only referring to fish.

- RHP stays for Resource Holding Potential and not for resource retention power.

- Line 753: hierarchy is not a property of a population

- Line 891-893: Yes, it is a promising avenue of research, but I think you need to say why and maybe provide a reference?

---

## Round 0.3 · accepted · Accept

Thank you for your attention to the remaining minor issues. I am happy to now accept your manuscript.